# A Cross-Sectional Evaluation of Caregiver Burden in Schizophrenia Care: Findings from Western Saudi Arabia with Policy Implications for Preventive Mental Healthcare

**DOI:** 10.3390/healthcare14010055

**Published:** 2025-12-25

**Authors:** Ashokkumar Thirunavukkarasu, Ebtehal Mobarak Zawawi

**Affiliations:** Department of Family and Community Medicine, College of Medicine, Jouf University, Sakaka 72388, Aljouf, Saudi Arabia; betty_1st@hotmail.com

**Keywords:** caregiver burden, policy implications, schizophrenia, Saudi Arabia, emotional strain

## Abstract

**Background/Objectives**: Research about the effects of schizophrenia, along with caregiver burden, exists extensively in Western countries. However, research on Middle Eastern societies, especially Saudi Arabia, is limited. We assessed the burden experienced by caregivers of individuals with schizophrenia and identified the associated factors contributing to it. **Methods**: The current cross-sectional study was conducted in Jeddah, Western Saudi Arabia, from December 2024 to March 2025. We used a validated Arabic data collection tool comprising 22 items that assessed five domains of caregiver burden. The associations between background characteristics and individual domains were determined by the Mann–Whitney U test and the Kruskal–Wallis test. We applied binomial regression analysis to find the factors associated with caregiver burden. **Results**: Of the 330 participants studied, no burden was observed in 17.9%. The remaining had mild (25.8%), moderate (34.8%), and severe (21.5%) burdens. Among the domains, emotional strain showed the highest mean (11.52 ± 4.32), followed closely by time and social limitations (11.29 ± 5.07) and health and financial impacts (11.08 ± 5.08). The caregiver burden was significantly higher among the adult children caring for their parents (*p* = 0.034) and lower among the participants working in the government sector (*p* = 0.022). **Conclusions:** The findings suggest a policy-relevant support program that includes workplace flexibility and financial help to manage their overall caregiving load and improve their health. Future research should explore the effectiveness of support strategies tailored to caregivers in different sociocultural contexts to enhance both caregiver and patient outcomes.

## 1. Introduction

Mental health disorders are among the most pressing public health concerns worldwide [1,2]. According to the World Health Organization (WHO), mental illnesses contribute significantly to the global burden of disease, affecting millions of individuals and their families [1]. Among these disorders, schizophrenia stands out as one of the most severe and disabling psychiatric conditions [3,4,5]. Schizophrenia is a chronic mental disorder characterized by problems in perception, language, and emotion, as well as life-related behaviors and performance. Schizophrenia is classified as a significant mental illness that causes considerable functional deficits and increases the likelihood of disability [6,7]. Over the past couple of decades, the incidence, prevalence, and disability due to schizophrenia have increased significantly worldwide [3].

People who have schizophrenia and other serious mental illnesses commonly rely on family members for primary treatment. The disorder not only affects individuals but also places a significant burden on caregivers who provide continuous emotional, physical, and financial support [8,9,10]. For example, a recent systematic review by Karambelas GJ et al. reported that caregivers play an important role in managing schizophrenia and bipolar disorder [11]. Caregivers are similar to individuals suffering from medical and mental ailments, as the vast majority of caregivers face symptoms such as anxiety disorders and depression, as well as a variety of economic and professional challenges. Because chronic diseases influence caregivers’ symptoms and behaviors, social actions directed at them create specific patient care demands [12,13]. Family members who assist and care for schizophrenia patients may confront difficulties due to their diminished capacities. This can result in changes to everyday routines, positions, and duties. According to Kaya and Öz, families with a member suffering from a mental disorder frequently face adjustment challenges [14]. Family caregivers are typically responsible for a wide range of tasks when caring for these patients. In addition to managing unexpected patient behaviors such as antagonism and aggression, they must perform several functions in their care, placing a significant strain on them. It has been discovered that a high proportion of family caregivers of schizophrenia patients carry a heavy load, which reduces their quality of life and exposes them to a variety of risks [15,16].

While schizophrenia and its impact on caregivers have been widely studied in Western countries, research in Middle Eastern societies, including Saudi Arabia, remains limited. Most of the caregiver burden studies are related to other diseases or overall physical and mental health [17,18]. Cultural norms, religious beliefs, and family structures influence caregiving experiences in ways that differ from Western contexts. In Saudi Arabia, mental health services continue to face structural limitations, including under-investment, workforce shortages, and fragmented service delivery across regions [19,20]. The societal stigma surrounding mental illness further complicates caregiving, as families may avoid seeking professional help due to fear of social judgment [21,22]. Moreover, deeply rooted cultural beliefs, such as attributing mental illness to supernatural causes or concerns about family shame, intensify caregiving challenges and place additional pressure on families providing care [17,23]. Understanding the unique challenges faced by caregivers in this region is essential to developing effective mental health strategies that cater to their needs. A systematic meta-synthesis of Middle Eastern studies supported these concepts [23]. They reported that family caregivers of people with severe mental illness experienced high emotional distress and burden, highlighting an urgent need for more support.

One study conducted in Saudi Arabia on the burdens faced by caregivers of people who have schizophrenia discovered that these people’s needs require a variety of responses, including educational training on effective coping strategies and psychological support in the form of counseling or group therapy [17]. Furthermore, one study sought to do a quantitative synthesis of the clinical correlates of caregiver burden in schizophrenia studies published over the previous two decades. The findings revealed a link between a higher load and disease-related risk factors such as more severe symptoms, more overall psychopathology, greater severity of functional impairment, and a longer duration of illness. The findings indicate that research parameters (such as study quality, participants, and location) moderate the correlations between these disease-related risk factors and carer burden. Psychosocial therapies are critical for family carers of individuals with schizophrenia [9]. In line with this, a recent meta-analysis found that psychoeducational family interventions significantly reduce caregiver burden and improve caregivers’ well-being [24]. Previous studies across different countries consistently report high levels of emotional distress, social disruption, and financial strain among caregivers of individuals with schizophrenia, indicating that the burden is multidimensional and persistent, irrespective of geographic or cultural setting [11,25,26].

Given the immense burden on caregivers, there is a growing need to implement preventive mental health strategies, including psychological support programs, caregiver training, community awareness campaigns, and policy reforms [27,28,29]. By identifying the specific stressors caregivers face, policymakers and healthcare providers can design interventions to alleviate caregiver burden and enhance the overall well-being of both patients and their families. Understanding these issues in the local context is essential, as evidence from Saudi Arabia remains limited and largely non-condition-specific, particularly regarding the multidimensional burden experienced by caregivers of individuals with schizophrenia. There is also a lack of recent data from the western region of the country, and no studies have examined how different sociodemographic or caregiving characteristics relate to specific burden domains. Hence, the present study aimed to assess the burden experienced by caregivers of individuals with schizophrenia in Jeddah and identify the associated factors contributing to it.

## 2. Materials and Methods

### 2.1. Study Description

The present cross-sectional study was conducted from December 2024 to March 2025. The study was carried out in mental health facilities in Jeddah, Saudi Arabia. These centers provide care and assistance to people suffering from mental illnesses, including schizophrenia, and are critical sites for studying caregiver load in this group. We included the family carers of people diagnosed with schizophrenia, aged 18 and above, who were actively involved in their care at mental healthcare centers in Jeddah, Saudi Arabia. We excluded caregivers of other mental health conditions and those who did not provide consent to participate in the study.

### 2.2. Sampling Procedures

We calculated the required number of caregivers using an online sample size calculator. During estimation, we considered 70% as the expected proportion of caregiver burden, a 95% confidence interval (CI), and a 5% margin of error. Given the scarcity of relevant studies in the Saudi context, we obtained the 70% expected proportion in the pilot study, which is a common method for determining the sample size [30,31]. Applying these values, we concluded that 325 participants were the minimum number of required caregivers for this study. To enhance transparency, the 70% estimate from the pilot study corresponds to an approximate 95% confidence interval of 62% to 77%, and this range was considered when determining the required sample size. The eligible participants were invited during the follow-up visits at the mental health centers using a convenience sampling method. This approach was chosen because caregivers of individuals with schizophrenia frequently attend scheduled follow-up visits, making this setting the most practical and feasible way to reach active caregivers who are directly involved in daily patient support.

### 2.3. Data Collection Steps

The researchers obtained ethical approval from the IRB, Ministry of Health, Jeddah (Approval no: A02056, Dated: 21 November 2024). Furthermore, we conducted this study in accordance with the Declaration of Helsinki. After being briefed on the study and obtaining informed consent, participants were asked to complete the Google form on the data collector’s electronic device. The authors used a validated Arabic version of the caregiver burden scale. The authors developed and validated this tool using standard protocols. Firstly, we developed the tool in English and translated it into Arabic using a translation-back translation process [32]. Initially, content validation was conducted by the experts from relevant specialties, including psychiatry, family medicine, public health, and nursing. The initial pool of items was generated based on established caregiver-burden frameworks, particularly the Zarit Burden Interview (ZBI-22), and relevant literature [33,34,35], and was further adapted to reflect sociocultural factors specific to the Saudi context to ensure conceptual relevance and applicability. The experts conducted a focus group discussion, and the content was discussed based on existing literature [34,36]. As part of the validation process, a pilot study was conducted among 40 eligible participants using the developed tool to assess its feasibility, reliability, and effectiveness in measuring caregiver burden. This step was aimed at identifying any ambiguities, evaluating the consistency of responses, and ensuring the tool’s applicability within the target population. All caregivers found the tool easy to comprehend and respond to. On average, the participants completed the survey in 10 min. The developed tool demonstrated good internal consistency, with a Cronbach’s alpha value of 0.85, indicating that the items reliably measure the intended construct and can be considered a dependable instrument for assessing caregiver burden. Hence, we performed the main study using the developed tool. Additionally, the KMO value (0.77) and a significant Bartlett’s test of sphericity (χ^2^ = 3032.44, df = 79, *p* < 0.001) confirmed the suitability of the data for factor analysis. The resulting factor loadings supporting the five-domain structure are provided in Appendix A. The EFA results demonstrated that the 22 items clustered coherently into the predefined five domains, providing empirical support for the scale’s structural validity and confirming that each domain represents a distinguishable aspect of caregiver burden.

The questionnaire has two sections. The first section asked about the caregiver’s sociodemographic characteristics, and the second section inquired about the caregiver’s burden. The sociodemographic section included variables commonly identified in the caregiver-burden literature as relevant predictors, including patients’ age, gender, and duration since diagnosis, as well as caregivers’ age, gender, education level, occupation, monthly income, perceived social support, and relationship to the patient. These variables were collected because they are theoretically grounded and consistently reported as potential confounders influencing caregiver burden in previous research. The caregiver burden scale (Arabic) had 22 items, each scored on a 5-point scale from 0 (never) to 4 (nearly always). The total score ranges from 0 to 88, with higher ratings suggesting a greater caregiver load. The overall score is interpreted as follows: 0–20 indicates little or no load, 21–40 indicates mild to moderate burden, 41–60 indicates moderate to severe burden, and 61–88 indicates severe burden. These cut-offs follow the classification widely used in studies applying the Zarit Burden Interview (ZBI-22), which uses the same scoring range and categorical thresholds [33,35,37]. Furthermore, the 22 items were categorized into five domains: Emotional Strain (Q1–Q5), assessing stress and frustration related to caregiving; Time and Social Limitations (Q6–Q10), evaluating the impact on personal time and social interactions; Interpersonal Relationships and Dependency (Q11–Q15), examining the effects on relationships and caregiver dependency; Health and Financial Impacts (Q16–Q20), addressing the caregiver’s physical well-being and financial strain; and Caregiving Expectations (Q21–Q22), exploring perceived pressure and future concerns regarding caregiving responsibilities.

### 2.4. Data Analysis

The authors used the Statistical Package for the Social Sciences version 21 (SPSS, V 21.0) (IBM Corp., Armonk, NY, USA) for data analysis. We presented background characteristics and other descriptive data, including frequencies and proportions. The gathered data did not meet the normality assumption. Hence, the association between background characteristics and individual domains was determined by the Mann–Whitney U test and the Kruskal–Wallis test. Furthermore, we applied binomial logistic regression analysis to identify the factors associated with the caregiver burden. All variables collected in the sociodemographic section were considered theoretically relevant for inclusion in the regression model and were therefore entered simultaneously to ensure appropriate adjustment for potential confounding. Multicollinearity was assessed using variance inflation factors (VIF), and no significant multicollinearity was observed. A *p*-value less than 0.05 was considered statistically significant.

## 3. Results

During the study period, we contacted 374 caregivers; 330 agreed to participate (response rate of 88.2%). Table 1 presents the background characteristics of caregivers and their patients. The majority of patients were aged between 20 and 40 years (42.7%) and were males (58.8%), and most had been diagnosed for 2–5 years (41.7%). Regarding caregivers, the majority were aged 40 to 50 years (41.8%), were female (54.2%), and worked in the private sector, within the income bracket of 5000 to 7000 SAR. Regarding relationships with patients, the majority were parents (26.4%) or spouses (24.8%).

Caregivers’ responses in each of the 22 items are given in Appendix A. Among respondents, the highest proportion of “nearly always” responses was observed for feeling emotionally strained (27.9%), followed by fear of being unable to continue caregiving in the future (26.4%), and feeling emotionally worn out from caregiving tasks (26.1%). We also observed that the highest levels of “never” responses were for feeling uneasy about how the patient’s behavior is perceived by others (15.8%), followed by perceiving that their patient depends on them more than necessary (15.2%), and feeling like they are the only person their relative relies on for help (14.8%).

Table 2 and Figure 1 present the descriptive statistics for caregiver burden domains. The total burden score had a mean of 49.18 (±17.33), with a median of 57 (IQR: 40–60). Among the domains, emotional strain showed the highest mean (11.52 ± 4.32), followed closely by time and social limitations (11.29 ± 5.07) and health and financial impacts (11.08 ± 5.08).

Regarding the association between caregiver burden domains and background characteristics, we found that most of the factors were not statistically significant across the domains, except caregiver age showed a significant association with the caregiving expectation domain (*p* = 0.011), with older caregivers reporting higher burden (higher mean rank). Another significant association was demonstrated between occupation with time and social limitations (*p* = 0.012), where retired caregivers reported the highest burden. No other variables, including patient and caregiver gender, education level, income, and perceived social support, have shown significant associations with caregiver burden domains (Table 3).

Of the 330 participants studied, no burden was observed in 17.9% of the participants. The remaining had mild (25.8%), moderate (34.8%), and severe (21.5%) burdens (Table 4).

The factors associated with caregiver burden that were identified through the binomial logistic regression analysis are depicted in Table 5. The caregiver burden was significantly lower among the participants working in the government sector (adjusted odds ratio [AOR] = 0.39, 95% confidence interval [CI] = 0.28–0.61, *p* = 0.022) and those with an income of more than 7000 SAR (AOR = 0.53, 95% CI = 0.31–0.70, *p* = 0.018). The caregiver burden was significantly higher among the adult children caring for their parents (AOR = 1.95, 95% CI = 1.05–3.60, *p* = 0.034).

## 4. Discussion

The present study assessed the burden experienced by caregivers of individuals with schizophrenia in Jeddah, Saudi Arabia. Among the identified caregiver burden domains, emotional stress appeared as the most substantial, while time and social restrictions, along with health and financial challenges, followed behind. Emotional distress experienced by caregivers creates multiple health problems, including depression, together with sleep disturbances and worsened physical health, which may lead to poor patient outcomes [38,39]. For instance, a study by Wang X et al. stated that hospital readmission was high among schizophrenic patients with a highly expressed emotion caregiver [40]. The constraints of caregiving work coupled with time restrictions, isolate caregivers socially, keeping them apart from their personal lives. Emotional distress and social limitations stand out as the main concerns that affect caregivers of schizophrenia patients, based on some Western studies [41,42]. Similarly, a systematic review of caregivers of persons with severe mental disorders reported substantial financial strain, underscoring the multidimensional nature of caregiver burden [43]. Our findings are supported by previous studies in the region that are related to the burden of mental health disorders on caregivers [17,44]. In fact, a systematic review of family caregivers in Middle Eastern countries found pervasive caregiver distress and an urgent call for expanded support and resources for these families [23].

More caregivers reported experiencing emotional strain, future caregiving concern, and emotional exhaustion than any other responses. The most common responses indicating never experiencing these concerns included questions about others judging the patient’s actions and feeling that the patient depends exclusively on the caregiver. Caregivers need specific psychological support because of their high levels of emotional exhaustion, together with their uncertainty regarding future caregiving durations [12,38,39,45]. The extent of emotional distress plays a central role in causing burnout among caregivers, alongside depression, and creating declines in caregiving quality, so targeted stress management programs become essential. Caregivers in this context show less worry about patient perceptions because stigma does not seem to present as much of an issue when compared to other communities that face mental health-related stigma. Research carried out in different nations, including China and India, displays stigma concerns and elevated emotional burdens on caregivers [46,47,48]. Family caregiving is strongly shaped by cultural and religious norms that emphasize close-knit family responsibility, which often places long-term care expectations on immediate relatives. These expectations, combined with the limited availability of community-based mental health services and structured caregiver support programs, may contribute to sustained caregiving demands despite lower perceived stigma [19,23,49].

Most background variables measured in this study were not significantly related to the burden experienced by caregivers. Caregivers tended to experience higher caregiving-expectation burdens when older than younger caregivers. Retired caregivers experienced the greatest burden in terms of time and social constraints among all groups in this study. Physical and psychological stress from caregiving appears more intense for older caregivers because of their declining health condition as they age. Similar to the present study, Phillips R et al. reported in 2022 that older people had a higher risk of suffering mental health issues due to caregiver responsibility [50]. However, some studies, including Phillips R et al., reported varying results related to other variables [39,50,51]. For example, a 2025 systematic review of Indian studies by Grover S et al. reported that higher age is associated with a lower burden among the carers of Schizophrenia patients [52].

The high importance of specialized help programs for elderly caregivers calls for effective interventions, including home health visits and breaks from caregiving. Retired caregivers experience a higher caregiver burden because caring responsibilities take over their lives after they stop working, hence indicating a strong need for community programs and social opportunities.

Most caregivers reported a burden at some point; those facing moderate burden were the most numerous, followed by those facing mild and severe burden. Evidence from this study establishes the urgent requirement for interventions to reduce mental strain, which afflicts many caregivers at a moderate to severe level. Health systems need to use schizophrenia management plans for incorporating specialized interventions that help caregivers, such as mental health counseling, stress management workshops, and caregiver support groups. Research has shown that caregivers of patients with schizophrenia endure a higher burden than parents taking care of people with other persistent medical conditions [53,54]. Patient caregivers in systems that provide structured training and community mental health programs experience much lower burden levels. Research data show that Saudi Arabia lacks supportive programs for caregivers while demanding immediate policy innovations to support their care. For instance, a systematic review of caregivers similarly found limited formal support and respite for family carers, which may contribute to elevated caregiver strain [23]. In contrast to the present study, Shamsaei F et al. from Iran reported higher levels of moderate and severe caregiver burden for patients with schizophrenia [55]. The observed differences in caregiver burden across studies may be attributed to variations in sociocultural contexts, caregiving expectations, and the assessment tools utilized.

The study revealed several factors associated with caregiver burden through logistic regression analysis. Caregivers working in the government sector and individuals earning higher incomes demonstrated substantially lower burden levels. Government-employed caregivers experienced less burden due to stable employment, health benefits, and structured work environments that minimize their caregiving stress levels. According to this research, financial stability significantly reduces stress in caregiving situations. Our study findings are supported by previous studies from different parts of the world, which have found that financial pressure is one of the critical associated factors of the increased caregiver burden [10,11,42]. Economic programs that support caregivers need to be implemented because lower-income families require financial benefits to support their caregiving duties. Our study’s findings propose viable policy directions for consideration in the Saudi context. Enhancing community-based mental health services may help reduce reliance on family caregivers. Setting up caregiver units at the mental health facilities to offer psychoeducation, respite activities, and special helplines can offer viable relief to the families. Flexibility in the workplace for employed caregivers, in line with national labor laws, and specific financial assistance to low-income households could further alleviate the burden of caregiving.

The authors conducted this research using a standardized questionnaire in a unique sociocultural setting. Nevertheless, we would like to mention some limitations of this research. Firstly, the study was carried out in mental health facilities in Jeddah, Saudi Arabia. Considering the huge variations in sociocultural settings across Saudi Arabia, the prevalence of caregiver burdens in this study may not be the same in other regions. Secondly, we included only the caregivers of schizophrenic patients. Therefore, the findings may differ from those of other mental health conditions. Next, as this was a cross-sectional study, causal or temporal relationships between the associated factors and caregiver burden cannot be established. The findings should therefore be interpreted as associations rather than causal effects. Furthermore, the use of convenience sampling may have influenced the representativeness of the sample, further limiting the external validity of the findings. Caregivers who attend follow-up clinics may differ from those providing care in community settings, which may lead to under- or over-estimation of caregiver burden. Finally, recall bias and selection bias must be kept in mind while reading this manuscript.

## 5. Conclusions

This study reveals that caregivers of people with schizophrenia endure significant burdens, particularly in emotional distress and problems with social activities. Caring for relatives with schizophrenia was associated with a higher burden on caregivers who had lower incomes, and adult children provided care. Simultaneously, government employment, coupled with financial stability, was associated with lower burden. The burden experienced by retired individuals, along with those in older age groups, was observed to be most prominent in particular caregiving domains, indicating a need for assistance programs. The study suggests that caregivers may benefit from structured support programs, workplace flexibility, and financial assistance to reduce their overall burden and improve their well-being. Future research should explore the effectiveness of support strategies tailored to caregivers in different sociocultural contexts to enhance both caregiver and patient outcomes.

## Figures and Tables

**Figure 1 healthcare-14-00055-f001:**
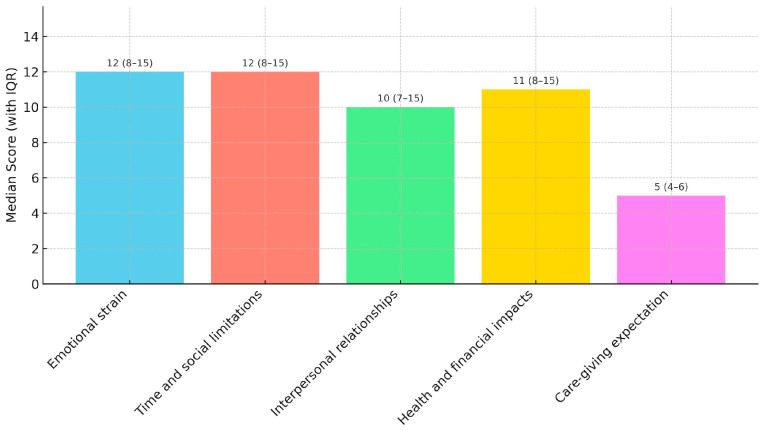
Median caregiver burden score with IQR values.

**Table 1 healthcare-14-00055-t001:** Background characteristics of the caregivers (n = 330).

Characteristics	Frequency	Percentage
Patients’ characteristics		
Age		
Less than 20 years	105	31.8
20 to 40 years	141	42.7
More than 40 years	84	25.5
Gender		
Female	136	41.2
Male	194	58.8
Duration since diagnosis		
Less than 2 years	101	30.6
2 to 5 years	136	41.2
More than 5 years	93	28.2
Caregivers’ characteristics		
Age		
Less than 40 years	110	33.3
40 to 50 years	138	41.8
More than 50 years	82	24.8
Gender		
Female	179	54.2
Male	151	45.8
Education level		
Up to high school	101	30.7
Bachelor’s degree	101	30.6
Postgraduate degree	128	38.7
Occupation		
Private	98	29.7
Government	130	39.4
Unemployed	64	19.4
Retired	38	11.5
Monthly income		
Less than 5000 SAR	100	30.3
5000 to 7000 SAR	134	40.6
More than 7000 SAR	96	29.1
Perceived social support		
No support	122	37.0
Some support	110	33.3
Enough support	98	29.7
Relationship with the patients		
Extended family (Uncles, aunts, etc.)	30	9.1
Siblings	54	16.4
Spouse	82	24.8
Adult children caring for parents	77	23.3
Parents	87	26.4

**Table 2 healthcare-14-00055-t002:** Descriptive Statistics of Caregiver Burden Domains (Mean, SD, Median, IQR).

Domains	Mean ± SD	Median (IQR)
Emotional strain	11.52 ± 4.3	12 (8–15)
Time and social limitations	11.29 ± 5.07	12 (8–15)
Interpersonal relationships and dependency	10.51 ± 4.95	10 (7–15)
Health and financial impacts	11.08 ± 5.08	11 (8–15)
Care-giving expectation	4.79 ± 2.14	5 (4–6)
Total	49.18 ± 17.33	57 (40–60)

**Table 3 healthcare-14-00055-t003:** Association Between Caregiver Burden Domains and Background Characteristics. Test applied: Mann–Whitney U and Kruskal–Wallis Tests.

Variables	Frequency	Emotional Strain		Time and Social Limitations		Interpersonal Relationships and Dependency		Health and Financial Impacts		Care Giving Expectation	
		Mean Rank	*p* Value	Mean Rank	*p* Value	Mean Rank	*p* Value	Mean Rank	*p* Value	Mean Rank	*p* Value
Patients’ characteristics											
Age			0.727		0.198		0.385		0.606		0.820
Less than 20 years	105	169.77		175.27		169.04		172.20		165.51	
20 to 40 years	141	160.70		154.72		157.47		159.98		162.44	
More than 40 years	84	168.21		171.38		174.55		166.39		170.62	
Gender			0.376		0.629		0.796		0.988		0.676
Female	136	159.96		168.53		163.89		165.40		162.91	
Male	194	169.38		163.38		166.63		165.57		167.32	
Duration since diagnosis			0.404		0.132		0.379		0.822		0.695
Less than 2 years	101	168.07		172.92		165.08		169.88		167.12	
2 to 5 years	136	157.57		152.98		158.45		162.07		160.53	
More than 5 years	93	174.30		175.76		176.27		165.76		171.01	
Caregivers’ characteristics											
Age			0.935		0.731		0.338		0.390		0.011
Less than 40 years	110	163.57		160.67		155.37		155.74		150.10	
40 to 50 years	138	167.74		165.69		167.69		168.47		168.68	
More than 50 years	82	164.33		171.66		171.66		173.59		180.82	
Gender			0.114		0.759		0.585		0.607		0.872
Female	179	173.10		166.97		162.87		167.97		164.73	
Male	151	156.49		163.75		168.61		162.57		166.41	
Education level			0.132		0.365		0.454		0.516		0.992
Up to high school	101	166.93		176.53		173.81		167.90		163.23	
Bachelor’s degree	101	174.33		162.56		162.56		156.28		165.22	
Postgraduate degree	128	152.71		157.44		157.44		167.76		166.60	
Occupation			0.756		0.012		0.222		0.279		0.516
Private	98	157.05		146.88		149.64		153.14		158.12	
Government	130	168.28		172.38		169.43		164.64		167.14	
Unemployed	64	168.18		167.06		171.95		182.30		162.14	
Retired	38	173.26		187.37		182.09		172.01		184.57	
Monthly income			0.507		0.388		0.176		0.158		0.138
Less than 5000 SAR	100	174.69		169.44		169.73		164.61		161.06	
5000 to 7000 SAR	134	160.72		156.97		154.17		155.71		157.40	
More than 7000 SAR	96	162.61		173.31		176.91		180.09		181.43	
Perceived social support			0.571		0.722		0.713		0.702		0.388
No support	122	160.15		160.29		171.01		170.98		172.13	
Some support	110	173.08		170.21		163.29		163.85		167.56	
Enough support	98	163.66		166.70		161.12		160.52		154.93	

**Table 4 healthcare-14-00055-t004:** Caregiver burden categories (n = 330).

Variable	Frequency	Percentage
No burden	59	17.9
Mild burden	85	25.8
Moderate burden	115	34.8
Severe burden	71	21.5

**Table 5 healthcare-14-00055-t005:** Factors associated with the caregiver burden. Test applied: Binomial logistic regression analysis (n = 330).

Variables	Total (n = 330)	No/Mild (n = 144)	Moderate/Severe (n = 186)	Adjusted OR (95% CI)	*p*-Value
Patients’ characteristics					
Age					
Less than 20 years	105	42	63	Ref	
20 to 40 years	141	68	73	1.21 (0.54–2.59)	0.789
More than 40 years	84	34	50	2.60 (0.38–1.11)	0.116
Gender					
Female	136	59	77	Ref	
Male	194	85	109	1.01 (0.65–1.58)	0.938
Duration since diagnosis					
Less than 2 years	101	41	60	Ref	
2 to 5 years	136	67	69	0.92 (0.52–1.64)	0.924
More than 5 years	93	36	57	0.65 (0.38–1.11)	0.650
Caregivers’ characteristics					
Age					
Less than 40 years	110	54	56	Ref	
40 to 50 years	138	57	81	0.70 (0.39–1.25)	0.224
51 and above years	82	33	49	0.96 (0.55–1.67)	0.877
Gender					
Female	179	77	102	Ref	
Male	151	67	84	1.06 (0.68–1.65)	0.805
Education level					
No formal education					
Up to high school	101	40	61	Ref	
Bachelor’s degree	101	42	59	1.29 (0.74–2.25)	0.372
Postgraduate degree	128	62	66	1.32 (0.78–2.23)	0.301
Occupation					
Private	98	50	48	Ref	
Government	130	57	73	0.39 (0.28–0.61)	0.022
Unemployed	64	26	38	0.52 (0.23–1.14)	0.103
Retired	38	11	27	0.59 (0.25–1.41)	0.238
Monthly income					
Less than 5000 SAR	100	39	61	Ref	
5000 to 7000 SAR	134	70	64	0.89 (0.50–1.60)	0.714
More than 7000 SAR	96	35	61	0.53 (0.31–0.70)	0.018
Social Support					
No support	122	51	71	Ref	
Some support	110	47	63	1.23 (0.72–2.10)	0.446
Enough support	98	46	52	1.19 (0.69–2.05)	0.542
Relationship with the patients					
Extended family (Uncles, aunts, etc.)	30	15	15	Ref	
Siblings	54	20	34	1.12 (0.49–2.57)	0.786
Spouse	82	30	52	1.91 (0.95–3.82)	0.068
Adult children caring for parents	77	33	44	1.95 (1.05–3.60)	0.034
Parents	87	46	41	1.49 (0.81–2.77)	0.201

## Data Availability

Data is contained within the article or Appendix A.

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
