# Peer review of "A Cross-Sectional Evaluation of Caregiver Burden in Schizophrenia Care: Findings from Western Saudi Arabia with Policy Implications for Preventive Mental Healthcare"

_healthcare, 2025, doi:10.3390/healthcare14010055_

Round 1
Reviewer 1 Report
Comments and Suggestions for Authors
I would like to thank you for your detailed review of the manuscript “A Cross-Sectional Evaluation of Caregiver Burden in Schizophrenia Care: Findings from Western Saudi Arabia with Policy Implications for Preventive Mental Healthcare,” submitted to Healthcare magazine.
First, I would like to give a general assessment. The manuscript addresses a relevant and under-explored topic in the Middle East region: the burden of caregiving in schizophrenia within the Saudi sociocultural context. The study is timely, contributes to the regional literature on mental health, and proposes practical implications for health policy. However, although the work has clear strengths in its basic design and contextual justification, the methodology, analytical development, and discussion could be strengthened to meet the standards of rigor and validity required by JCR journals.
First, I would like to highlight the positive aspects. Social and health relevance: it addresses an emerging public problem in Saudi Arabia, with implications for preventive mental health policies. Cultural contextualization is adequate: it highlights the cultural, religious, and family characteristics that influence the role of the caregiver. In addition, the use of a validated instrument: the development and validation of an Arabic version of the caregiver burden instrument is a useful contribution. The statistical analysis is technically correct: the use of nonparametric tests (Mann–Whitney, Kruskal–Wallis) and binomial regression is appropriate for non-normal data. The discussion is coherent and includes recent references (2022–2024).
Aspects to be improved
- Methodology
- Cross-sectional design: Limits the ability to establish causal or temporal relationships between predictor variables and caregiver burden. It is suggested that the results describe associations, not causal relationships.
- Convenience sampling: Reduces representativeness and may introduce selection bias. It is recommended that this limitation and its impact on external validity be discussed more explicitly.
- Instrument validation: Although Cronbach's α = 0.85 is indicated, it would be important to also report construct validity (e.g., exploratory or confirmatory factor analysis), and no comparison with established scales (such as the Zarit Burden Interview) is presented, which would limit international comparability.
- Control of confounding factors: The binomial regression seems to include only a subset of variables. It is suggested to justify the selection of covariates and explore more comprehensive multivariate models (age, sex, income, social support).
- Sample size: Although it is calculated appropriately, the estimate based on a value of 70% from the pilot study introduces uncertainty, and it would be desirable to include the expected confidence interval to reinforce methodological transparency.
- Analysis and interpretation of results: The descriptive results are clear, but the discussion does not delve into the possible interaction between variables (e.g., between income level and social support). It would be useful to include a multivariate analysis adjusted for possible confounding factors and discuss the marginal effects. It is also recommended to include additional tables or figures that visually show the most affected domains of burden.
- Conclusions: The conclusions are consistent with the data, but should be formulated with greater caution, avoiding causal implications (“X predicts Y”) and replacing them with associations (“X is associated with greater burden”). They could benefit from more critical reflection on how the findings could guide evidence-based prevention programs or psychosocial interventions.
Soundness and validity of conclusions
Given the methodology used (cross-sectional, with non-probabilistic sampling and no robust validation of the instrument), the conclusions are moderately valid in descriptive terms, but cannot be considered confirmatory or generalizable to the entire population of Saudi caregivers. The study provides useful local evidence, but requires caution in inference and more comparative empirical support to substantiate policy recommendations.
Suggestions for strengthening the theoretical and contextual review
The manuscript cites current literature, but could be enriched with recent systematic reviews on the role of the caregiver, stress, and psychosocial well-being, as well as on assessment tools for person-centered care and resilience.
It would be advisable to incorporate citations from systematic reviews that address these aspects. These citations would not only strengthen the theoretical framework but also increase the impact and scientific relevance of the article, integrating it into the international literature on caregiver burden and person-centered care.
Final recommendation
Suggested decision: Major Revision
The article is promising and relevant, but to meet the standards for publication in a JCR journal, it should:
- Delve deeper into the validation of the instrument and justify its choice over classic scales.
- Strengthen the statistical analysis with a multivariate model and control for confounders.
- Expand the theoretical framework with references to systematic reviews on the role of the caregiver, stress, resilience, and person-centered care.
- Reformulate the conclusions in a more cautious manner based on the limitations of the cross-sectional design.
Author Response
Authors’ reply/modifications according to the reviewer 1 comments/suggestions
General:
The authors would like to thank the reviewer for the precious time spent reviewing the paper and his excellent suggestions for improving it. Efforts have been made to modify the paper as per the reviewer’s suggestions and recommendations. The authors will be happy to hear a positive reply. All the points included according to the reviewer’s comments can be seen in track changes.
Specific response to the reviewer’s suggestions:
Kindly find the attached response to each question one by one:
Point 1: First, I would like to give a general assessment. The manuscript addresses a relevant and under-explored topic in the Middle East region: the burden of caregiving in schizophrenia within the Saudi sociocultural context. The study is timely, contributes to the regional literature on mental health, and proposes practical implications for health policy. However, although the work has clear strengths in its basic design and contextual justification, the methodology, analytical development, and discussion could be strengthened to meet the standards of rigor and validity required by JCR journals.
Response 1: Thanks for the comment. The authors are pleased to hear positive comments from the reviewer. We made the necessary changes according to the reviewer’s suggestions.
Point 2: First, I would like to highlight the positive aspects. Social and health relevance: it addresses an emerging public problem in Saudi Arabia, with implications for preventive mental health policies. Cultural contextualization is adequate: it highlights the cultural, religious, and family characteristics that influence the role of the caregiver. In addition, the use of a validated instrument: the development and validation of an Arabic version of the caregiver burden instrument is a useful contribution. The statistical analysis is technically correct: the use of nonparametric tests (Mann–Whitney, Kruskal–Wallis) and binomial regression is appropriate for non-normal data. The discussion is coherent and includes recent references (2022–2024).
Response 2: Thanks for the comment. The authors are pleased to hear positive comments from the reviewer. The reviewer’s positive feedback is highly motivating and supports our continued efforts to advance research in this important area.
Point 3: Cross-sectional design: Limits the ability to establish causal or temporal relationships between predictor variables and caregiver burden. It is suggested that the results describe associations, not causal relationships.
Response 3: Thanks for the comment. The authors agree that the cross-sectional design limits the ability to establish causality or temporal relationships between the independent variables and caregiver burden. According to the reviewer’s suggestion, we have revised the manuscript to ensure that all interpretations are framed as associations rather than causal effects. Furthermore, the words determinants/predictors are changed to “associated factors” in the revised manuscript.
Point 4: Convenience sampling: Reduces representativeness and may introduce selection bias. It is recommended that this limitation and its impact on external validity be discussed more explicitly.
Response 4: Thanks for the comment. According to the reviewer’s comments, we have enhanced the limitations of the study related to the sampling and external validity.
Point 5: Instrument validation: Although Cronbach's α = 0.85 is indicated, it would be important to also report construct validity (e.g., exploratory or confirmatory factor analysis), and no comparison with established scales (such as the Zarit Burden Interview) is presented, which would limit international comparability.
Response 5: Thanks for the comment. The authors acknowledge the importance of construct validation to be mentioned for strengthening the domain-specific results. According to the reviewer’s comments, in the revised manuscript, we have now included details of the Exploratory Factor Analysis (EFA), including KMO, Bartlett’s test, and the factor-loading results, which empirically support the five-domain structure of the 22-item scale. According to the reviewer’s comments, we have added clarification that item development was guided by established caregiver-burden frameworks, particularly the Zarit Burden Interview (ZBI-22), with the relevant references, ensuring conceptual consistency and enhancing replicability.
Point 6: Control of confounding factors: The binomial regression seems to include only a subset of variables. It is suggested to justify the selection of covariates and explore more comprehensive multivariate models (age, sex, income, social support).
Response 6: Thanks for the comment. The authors would like to clarify that we used the enter method in the logistic regression analysis. The model adjusted for patients’ age, gender, and duration since diagnosis, as well as caregivers’ age, gender, education level, occupation, monthly income, perceived social support, and relationship to the patient. According to the reviewer’s suggestions, this clarification has now been added to the Statistical Analysis section.
Point 7: Sample size: Although it is calculated appropriately, the estimate based on a value of 70% from the pilot study introduces uncertainty, and it would be desirable to include the expected confidence interval to reinforce methodological transparency.
Response 7: Thanks for the comment. The authors are pleased to hear the positive comments on the sample size. According to the reviewer’s suggestions, we revised it in the manuscript.
Point 8: Analysis and interpretation of results: The descriptive results are clear, but the discussion does not delve into the possible interaction between variables (e.g., between income level and social support). It would be useful to include a multivariate analysis adjusted for possible confounding factors and discuss the marginal effects. It is also recommended to include additional tables or figures that visually show the most affected domains of burden.
Response 8: Thanks for the comment. The authors conducted multivariable analysis (logistic regression) to adjust for covariates (variables of the study). The model adjusted for patients’ age, gender, and duration since diagnosis, as well as caregivers’ age, gender, education level, occupation, monthly income, perceived social support, and relationship to the patient. According to the reviewer’s suggestions, this clarification has now been added to the Statistical Analysis section. Following the reviewer’s suggestions, we have included a new figure (Figure 1).
Point 9: Conclusions: The conclusions are consistent with the data, but should be formulated with greater caution, avoiding causal implications (“X predicts Y”) and replacing them with associations (“X is associated with greater burden”). They could benefit from more critical reflection on how the findings could guide evidence-based prevention programs or psychosocial interventions.
Response 9: Thanks for the comment. According to the reviewer’s suggestions, we revised the conclusion section of the abstract and the main text.
Point 10: Given the methodology used (cross-sectional, with non-probabilistic sampling and no robust validation of the instrument), the conclusions are moderately valid in descriptive terms, but cannot be considered confirmatory or generalizable to the entire population of Saudi caregivers. The study provides useful local evidence, but requires caution in inference and more comparative empirical support to substantiate policy recommendations.
Response 10: Thanks for the comment. We agree with the reviewer’s comments. Accordingly, we revised the limitations and conclusions to address these comments.
Point 11: The manuscript cites current literature, but could be enriched with recent systematic reviews on the role of the caregiver, stress, and psychosocial well-being, as well as on assessment tools for person-centered care and resilience. It would be advisable to incorporate citations from systematic reviews that address these aspects. These citations would not only strengthen the theoretical framework but also increase the impact and scientific relevance of the article, integrating it into the international literature on caregiver burden and person-centered care.
Response 11: Thanks for the comment. According to the reviewer’s suggestions, we included systematic reviews from both the Middle East and International in the introduction and discussion. The original cited article number was 37. Now the revised manuscript cited article number is 60. Several of these newly included citations are systematic reviews.
Point 12: Delve deeper into the validation of the instrument and justify its choice over classic scales.
Response 12: Thanks for the comment. According to the reviewer, more details are included in the revised manuscript about the validation. We added a new supplementary Table (2) with the EFA analysis report. Furthermore, the justifications for the present tool over the classic tool are also included in the methods section of the revised manuscript.
Point 13: Strengthen the statistical analysis with a multivariate model and control for confounders.
Response 13: Thanks for the comment. Please find the details in response 6 and response 8.
Point 14: Expand the theoretical framework with references to systematic reviews on the role of the caregiver, stress, resilience, and person-centered care.
Response 14: Thanks for the comment. Please find the response no.11.
Point 15: Reformulate the conclusions in a more cautious manner based on the limitations of the cross-sectional design.
Response 15: Thanks for the comment. According to the reviewer’s suggestions, we revised the conclusion based on the limitations of the present study.
The authors thank the reviewer once again for the positive and constructive comments.

Reviewer 2 Report
Comments and Suggestions for Authors
This assessed the burden experienced by caregivers of individuals with schizophrenia and identified the associated factors contributing to it. This topic is important and interesting. Some comments are provided for improving the manuscript quality.
- The novelty and significance of the work should be clearly explained.
- What are the previous studies of carer burden?
- What is the sampling method? What are the inclusion and exclusion criteria for participant recruitment?
- What are the factors considered to influence the burden?
- How to categorize the burden into no burden, mild burden, moderate burden and severe burden?
- What are the practical implications of the work?
Author Response
Authors’ reply/modifications according to the reviewer 2 comments/suggestions
General:
The authors would like to thank the reviewer for the precious time spent reviewing the paper and his excellent suggestions for improving it. Efforts have been made to modify the paper as per the reviewer’s suggestions and recommendations. The authors will be happy to hear a positive reply. All the points included according to the reviewer’s comments can be seen in track changes.
Specific response to the reviewer’s suggestions:
Kindly find the attached response to each question one by one:
Point 1: This assessed the burden experienced by caregivers of individuals with schizophrenia and identified the associated factors contributing to it. This topic is important and interesting. Some comments are provided for improving the manuscript quality.
Response 1: Thanks for the comments. The authors are pleased to hear positive comments from the reviewer. We made changes in the revised manuscript as per the reviewer’s suggestions.
Point 2: The novelty and significance of the work should be clearly explained.
Response 2: Thanks for the comments. According to the reviewer’s suggestions, we revised the rationale and significance/need for the study in the revised manuscript.
Point 3: What are the previous studies of carer burden?.
Response 3: Thanks for the comments. According to the reviewer’s suggestions, we added a brief summary of findings from previous studies in the Introduction to clarify the established evidence on caregivers with new references. Furthermore, we included several systematic reviews to support our contextualization for the study. These studies consistently report multidimensional burden, including emotional distress, social limitations, and financial strain, among caregivers of individuals with schizophrenia.
Point 4: What is the sampling method? What are the inclusion and exclusion criteria for participant recruitment?
Response 4: Thanks for the comments. We included the family carers of people diagnosed with schizophrenia, aged 18 and above, who were actively involved in their care at mental healthcare centers in Jeddah, Saudi Arabia. We excluded caregivers of other mental health conditions and those who did not provide consent to participate in the study. The eligible participants were invited during the follow-up visits at the mental health centers using a convenience sampling method. This approach was chosen because caregivers of individuals with schizophrenia frequently attend scheduled follow-up visits, making this setting the most practical and feasible way to reach active caregivers who are directly involved in daily patient support. The details are included in the revised manuscript as per the reviewer’s suggestions.
Point 5: What are the factors considered to influence the burden?.
Response 5:Thanks for the comments. The authors considered patients’ age, gender, and duration since diagnosis, as well as caregivers’ age, gender, education level, occupation, monthly income, perceived social support, and relationship to the patient. These variables were selected based on previous literature identifying commonly reported determinants of caregiver burden, as well as their relevance to the sociocultural caregiving context in Saudi Arabia. Furthermore, these factors were considered for logistic regression model (enter method). According to the reviewer’s suggestions, this clarification has now been added to the methods section.
Point 6: How to categorize the burden into no burden, mild burden, moderate burden and severe burden?
Response 6: Thanks for the comment. The caregiver burden is categorized according to overall scores (0-20, 21-40, 41-60, and 61-88). These cut-offs follow the classification widely used in studies applying the Zarit Burden Interview (ZBI-22), which uses the same scoring range and categorical thresholds. This categorization approach is well-established in caregiver burden research, and we have now clarified it in the Methods section with the new references.
Point 7: What are the practical implications of the work?.
Response 7: Thanks for the comment. The study has numerous policy implications. According to the reviewer’s suggestions, we have enhanced policy implications in the discussion and conclusion sections.
The authors thank the reviewer once again for the positive and constructive comments.

Reviewer 3 Report
Comments and Suggestions for Authors
Overall Evaluation
This manuscript addresses an important topic by examining caregiver burden among family members of individuals with schizophrenia in western Saudi Arabia. The study provides meaningful descriptive insights; however, several major issues substantially limit the scientific rigor and interpretive validity of the work. These include the absence of construct validation for the measurement tool, the use of convenience sampling with limited external validity, insufficient justification for questionnaire development, overgeneralized policy implications, and several unreferenced or unsupported claims. Significant revisions are required to enhance the methodological robustness, theoretical grounding, and interpretative clarity of the manuscript.
1. Introduction
The introduction effectively highlights the global relevance of mental health disorders and the particular burden of schizophrenia. However, several key claims require clearer sourcing and more rigorous justification. For example, the manuscript states that “schizophrenia stands out as one of the most severe and disabling psychiatric conditions” (lines 40–44), yet no direct citation is provided for this statement beyond general WHO references. Such assertions should be supported with specific empirical sources.
Similarly, the passage describing preventive strategies—“These strategies can include psychological support programs, caregiver training, community awareness campaigns, and policy reforms…” (lines 89–90)—presents a list of interventions without citation. Since these form part of the study’s rationale, proper referencing is essential.
Additionally, while the authors mention cultural norms, religious beliefs, and stigma as relevant contextual factors in Middle Eastern societies (lines 67–70), these descriptions are brief and not sufficiently elaborated to situate the study within the sociocultural specificities of western Saudi Arabia. Given that the manuscript title emphasizes policy implications, the introduction should more clearly articulate the structural challenges in the Saudi mental healthcare system and the policy gaps this study intends to address.
2. Materials and Methods
The study design and general data collection procedures are described clearly; however, several methodological limitations must be addressed.
First, the most critical issue concerns the lack of construct validation for the 22-item caregiver burden scale. The manuscript states that the tool was developed, translated, reviewed by experts, and piloted (lines 125–137), and reports its internal consistency (Cronbach’s α = 0.85). However, no factor analysis or other structural validation is presented to justify the five-domain structure used in the analyses (lines 147–154). Without evidence of factorial validity, the interpretation of domain-specific results is significantly weakened.
Second, although the authors claim the tool was developed “based on existing literature” (line 130), the manuscript does not identify which established scales—such as the Zarit Burden Interview—or which theoretical foundations guided item selection. This lack of justification limits the transparency and replicability of the instrument.
Third, the authors recruited participants using convenience sampling during follow-up visits at mental health centers (lines 118–119). This approach constrains external validity, as caregivers who accompany patients to clinics may systematically differ from community-based caregivers. While some limitations are acknowledged later (lines 294–299), the methodological implications are not adequately addressed.
Finally, in the logistic regression analyses, the manuscript does not describe how predictor variables were selected, whether confounders were adjusted for, or whether multicollinearity was examined. Such details are necessary for evaluating the robustness of the statistical models.
3. Results
The results are generally well structured and clearly presented. Tables are readable, and the distribution of caregiver burden categories is documented appropriately.
However, interpretation of domain-level results remains inherently uncertain because the underlying factor structure of the measurement tool has not been validated. Without evidence supporting the five-domain model, differences in mean scores or rank-order comparisons may not reflect true underlying constructs.
Furthermore, while logistic regression outputs (Table 5) identify several statistically significant associations, the absence of model diagnostics (e.g., goodness-of-fit indices, multicollinearity statistics) prevents further assessment of the model’s adequacy. The manuscript also does not contextualize the clinical meaning of burden categories (e.g., moderate vs. severe burden), which would add interpretive depth.
4. Discussion and Conclusion
The discussion successfully links some findings to previous research. However, several issues require substantial refinement.
The manuscript asserts that “schizophrenia caregivers endure a higher burden than parents taking care of people with other persistent medical conditions” (lines 269–272), but does not cite specific empirical work supporting this cross-condition comparison. This is a major concern, given that the claim is used to justify policy recommendations.
Regional specificity is also limited. Although the discussion briefly notes that stigma appears less prominent in this sample compared to China or India (lines 243–246), it does not meaningfully explore sociocultural dynamics in western Saudi Arabia, such as family caregiving expectations, norms around mental illness, or differential access to community-based services.
The most significant weakness is the lack of concrete policy implications. Despite the title highlighting policy relevance, recommendations such as “structured support programs,” “workplace flexibility,” and “financial assistance” (lines 307–309) remain general and lack actionable detail. To justify the prominence of “policy implications” in the title, the discussion should incorporate specific proposals grounded in Saudi Arabia’s mental health system, resource structures, and cultural context.
Additionally, the manuscript frequently uses terms such as “predictors” and “determinants” (e.g., lines 279–284) despite using a cross-sectional design, which cannot establish temporal or causal relationships. The discussion and conclusion should use more cautious language, such as “associated factors.”
5. Other Remarks
The supplementary dataset is mentioned (line 312), but the manuscript does not explain how it links to the analyses presented, nor does it provide a clear codebook or mapping of items to domains. Enhancing transparency would facilitate replication.
A minor typographical error appears in Table 2: “Emotional stain” should be corrected to “Emotional strain.”
The paper would also benefit from referencing additional comparative literature from the Gulf region, as this would strengthen the regional contribution of the study.
Comments on the Quality of English LanguageThe manuscript is generally understandable, but several sentences are repetitive or overly general. Greater precision is needed to avoid implying causal relationships. Additionally, transitions between paragraphs could be improved for smoother flow. Moderate language editing is recommended to improve clarity, reduce redundancy, and enhance scholarly tone.
Author Response
Authors’ reply/modifications according to the reviewer 3 comments/suggestions
General:
The authors would like to thank the reviewer for the precious time spent reviewing the paper and his excellent suggestions for improving it. Efforts have been made to modify the paper as per the reviewer’s suggestions and recommendations. The authors will be happy to hear a positive reply. All the points included according to the reviewer’s comments can be seen in track changes.
Specific response to the reviewer’s suggestions:
Kindly find the attached response to each question one by one:
Point 1: This manuscript addresses an important topic by examining caregiver burden among family members of individuals with schizophrenia in western Saudi Arabia. The study provides meaningful descriptive insights; however, several major issues substantially limit the scientific rigor and interpretive validity of the work. These include the absence of construct validation for the measurement tool, the use of convenience sampling with limited external validity, insufficient justification for questionnaire development, overgeneralized policy implications, and several unreferenced or unsupported claims. Significant revisions are required to enhance the methodological robustness, theoretical grounding, and interpretative clarity of the manuscript.
Response 1: Thanks for the comments. The authors are pleased to hear the reviewer's positive comments on the significance of the topic. We revised the manuscript in accordance with the reviewer’s suggestions.
Point 2:The introduction effectively highlights the global relevance of mental health disorders and the particular burden of schizophrenia. However, several key claims require clearer sourcing and more rigorous justification. For example, the manuscript states that “schizophrenia stands out as one of the most severe and disabling psychiatric conditions” (lines 40–44), yet no direct citation is provided for this statement beyond general WHO references. Such assertions should be supported with specific empirical sources.
Response 2: Thanks for the comments. According to the reviewer’s suggestions, we have included new references for the justification of the statement.
Point 3: Similarly, the passage describing preventive strategies—“These strategies can include psychological support programs, caregiver training, community awareness campaigns, and policy reforms…” (lines 89–90)—presents a list of interventions without citation. Since these form part of the study’s rationale, proper referencing is essential.
Response 3: Thanks for the comments. According to the reviewer’s suggestions, we included references to support our rationale.
Point 4: Additionally, while the authors mention cultural norms, religious beliefs, and stigma as relevant contextual factors in Middle Eastern societies (lines 67–70), these descriptions are brief and not sufficiently elaborated to situate the study within the sociocultural specificities of western Saudi Arabia. Given that the manuscript title emphasizes policy implications, the introduction should more clearly articulate the structural challenges in the Saudi mental healthcare system and the policy gaps this study intends to address.
Response 4: Thanks for the comments. According to the reviewer’s suggestions, we have enhanced the introduction related to these aspects with additional references.
Point 5: The study design and general data collection procedures are described clearly; however, several methodological limitations must be addressed. First, the most critical issue concerns the lack of construct validation for the 22-item caregiver burden scale. The manuscript states that the tool was developed, translated, reviewed by experts, and piloted (lines 125–137), and reports its internal consistency (Cronbach’s α = 0.85). However, no factor analysis or other structural validation is presented to justify the five-domain structure used in the analyses (lines 147–154). Without evidence of factorial validity, the interpretation of domain-specific results is significantly weakened.
Response 5:Thanks for the comments. The authors acknowledge the importance of structural validation to be mentioned for strengthening the domain-specific results. According to the reviewer’s comments, in the revised manuscript, we have now included details of the Exploratory Factor Analysis (EFA), including KMO, Bartlett’s test, and the factor-loading results, which empirically support the five-domain structure of the 22-item scale.
Point 6: Second, although the authors claim the tool was developed “based on existing literature” (line 130), the manuscript does not identify which established scales—such as the Zarit Burden Interview—or which theoretical foundations guided item selection. This lack of justification limits the transparency and replicability of the instrument.
Response 6: Thanks for the comment. According to the reviewer’s comments, we have added clarification that item development was guided by established caregiver-burden frameworks, particularly the Zarit Burden Interview (ZBI-22), with the relevant references, ensuring conceptual consistency and enhancing replicability.
Point 7: Third, the authors recruited participants using convenience sampling during follow-up visits at mental health centers (lines 118–119). This approach constrains external validity, as caregivers who accompany patients to clinics may systematically differ from community-based caregivers. While some limitations are acknowledged later (lines 294–299), the methodological implications are not adequately addressed.
Response 7: Thanks for the comment. According to the reviewer’s suggestions and to address this issue, we have expanded the limitations section to explicitly acknowledge this potential source of selection bias and its implications for generalizability.
Point 8: Finally, in the logistic regression analyses, the manuscript does not describe how predictor variables were selected, whether confounders were adjusted for, or whether multicollinearity was examined. Such details are necessary for evaluating the robustness of the statistical models.
Response 8: Thanks for the comment. According to the reviewer’s comments, the necessary details are included in the revised manuscript’s statistical analysis section.
Point 9: The results are generally well structured and clearly presented. Tables are readable, and the distribution of caregiver burden categories is documented appropriately.
Response 9: Thanks for the comment. The authors are pleased to hear the positive comments from the reviewer regarding the results.
Point 10: However, interpretation of domain-level results remains inherently uncertain because the underlying factor structure of the measurement tool has not been validated. Without evidence supporting the five-domain model, differences in mean scores or rank-order comparisons may not reflect true underlying constructs.
Response 10: Thanks for the comment. According to the reviewer’s comments, in the revised manuscript, we have now included the results of an Exploratory Factor Analysis (EFA), including KMO and Bartlett statistics, and have provided the detailed factor-loading matrix in Supplementary Table 2. The EFA confirmed that the 22 items coherently clustered into the five predefined domains, thereby supporting the structural validity of the scale used in this study.
Point 11: Furthermore, while logistic regression outputs (Table 5) identify several statistically significant associations, the absence of model diagnostics (e.g., goodness-of-fit indices, multicollinearity statistics) prevents further assessment of the model’s adequacy. The manuscript also does not contextualize the clinical meaning of burden categories (e.g., moderate vs. severe burden), which would add interpretive depth.
Response 11: Thanks for the comment. According to the reviewer’s comment, in the revised manuscript, we have added clarification that standard diagnostic checks were performed. Specifically, multicollinearity was assessed using variance inflation factors (VIF), with no concerns identified, and overall model fit was reviewed using standard logistic regression diagnostic procedures. We agree that in the original manuscript, there were no clarifications about the reasons behind the categorization. Nonetheless, in the revised manuscript, we clarified it, i.e., we used similar to widely applied ZBI-22 scoring thresholds, which are based on established literature and provide meaningful distinctions between mild, moderate, and severe burden levels.
Point 12: The discussion successfully links some findings to previous research. However, several issues require substantial refinement.
Response 12: Thanks for the comment. According to the reviewer’s suggestions, we have revised the entire discussion.
Point 13: The manuscript asserts that “schizophrenia caregivers endure a higher burden than parents taking care of people with other persistent medical conditions” (lines 269–272), but does not cite specific empirical work supporting this cross-condition comparison. This is a major concern, given that the claim is used to justify policy recommendations.
Response 13: Thanks for the comment. According to the reviewer’s suggestions, we included empirical studies that supports this cross-condition comparision. This revision ensures that the statement is evidence-based and more cautiously worded.
Point 14: Regional specificity is also limited. Although the discussion briefly notes that stigma appears less prominent in this sample compared to China or India (lines 243–246), it does not meaningfully explore sociocultural dynamics in western Saudi Arabia, such as family caregiving expectations, norms around mental illness, or differential access to community-based services.
Response 14: Thanks for the comment. According to the reviewer’s comments, we have enhanced the discussion with the relevant references in the revised manuscript.
The following details were added:
Family caregiving is strongly shaped by cultural and religious norms that emphasize close-knit family responsibility, which often places long-term care expectations on immediate relatives. These expectations, combined with the limited availability of community-based mental health services and structured caregiver support programs, may contribute to sustained caregiving demands despite lower perceived stigma [19,23,52].
Point 15: The most significant weakness is the lack of concrete policy implications. Despite the title highlighting policy relevance, recommendations such as “structured support programs,” “workplace flexibility,” and “financial assistance” (lines 307–309) remain general and lack actionable detail. To justify the prominence of “policy implications” in the title, the discussion should incorporate specific proposals grounded in Saudi Arabia’s mental health system, resource structures, and cultural context.
Response 15: Thanks for the comment. According to the reviewer’s comments, the following details are included in the revised manuscript.
“Our study's findings propose viable policy directions for consideration in the Saudi context. Enhancing community-based mental health services may help reduce reliance on family caregivers. Setting up caregiver units at the mental health facilities to offer psychoeducation, respite activities, and special helplines can offer viable relief to the families. Flexibility in the workplace for employed caregivers, in line with national labor laws, and specific financial assistance to low-income households could further alleviate the burden of caregiving.”
Point 16: Additionally, the manuscript frequently uses terms such as “predictors” and “determinants” (e.g., lines 279–284) despite using a cross-sectional design, which cannot establish temporal or causal relationships. The discussion and conclusion should use more cautious language, such as “associated factors.”
Response 16: Thanks for the comment. According to the reviewer’s comments, we have carefully revised the manuscript to use the terms (predictors/determinants) cautiously, given the cross-sectional study design. These words are replaced with associated factors in the revised manuscript.
Point 17:The supplementary dataset is mentioned (line 312), but the manuscript does not explain how it links to the analyses presented, nor does it provide a clear codebook or mapping of items to domains. Enhancing transparency would facilitate replication.
Response 17: Thanks for the comment. As per the reviewer’s comments, we have uploaded as a supplementary file (zip file) titled data during resubmission
Point 18: A minor typographical error appears in Table 2: “Emotional stain” should be corrected to “Emotional strain..
Response 18: Thanks for the comment and notifying us of the error. According to the reviewer’s comments, we corrected it in the revised manuscript.
Point 19: The paper would also benefit from referencing additional comparative literature from the Gulf region, as this would strengthen the regional contribution of the study.
Response 19: Thanks for the comment. According to the reviewer’s suggestions, we have included studies from the Gulf region, including a recent systematic review, to strengthen the regional contribution of the study.
Point 20: The manuscript is generally understandable, but several sentences are repetitive or overly general. Greater precision is needed to avoid implying causal relationships. Additionally, transitions between paragraphs could be improved for smoother flow. Moderate language editing is recommended to improve clarity, reduce redundancy, and enhance scholarly tone.
Response 20: Thanks for the comment. According to the reviewer’s comments, the manuscript has undergone careful language revision to improve clarity, precision, and scholarly tone. Repetitive or overly general statements were removed, transitional flow between paragraphs was improved, and wording that could imply causal interpretation was corrected to reflect associative relationships consistent with the study design. The revised version has also been reviewed by a native English speaker to enhance readability.
The authors thank the reviewer once again for the positive and constructive comments.

Round 2
Reviewer 1 Report
Comments and Suggestions for Authors
The authors have effectively addressed most of the comments, but there are two points (11 and 14) that are related and concern me. Addressing them correctly will help improve the manuscript.
Both refer to citation 37 in the previous manuscript, which is now citation 60. I would like to point out that citation 60 does not exist in the updated version of the manuscript, but rather citation 11.
Furthermore, I would like to insist that the authors should enrich the manuscript with recent systematic reviews on the role of the caregiver, stress, and psychosocial well-being, as well as assessment tools for person-centered care and resilience. This would greatly help to improve the quality of the manuscript's content.
Point 11: The manuscript cites current literature, but it could be enriched with recent systematic reviews on the role of the caregiver, stress, and psychosocial well-being, as well as assessment tools for person-centered care and resilience. It would be advisable to incorporate citations from systematic reviews that address these aspects. These citations would not only reinforce the theoretical framework, but also increase the impact and scientific relevance of the article, integrating it into the international literature on caregiver burden and person-centered care.
Response 11: Thank you for your comment. In accordance with the reviewer's suggestions, we have included systematic reviews from both the Middle East and internationally in the introduction and discussion. The number of articles originally cited was 37. Now, the number of articles cited in the revised manuscript is 60. Several of these newly included citations are systematic reviews.
Point 14: Expand the theoretical framework with references to systematic reviews on the role of the caregiver, stress, resilience, and person-centered care.
Response 14: Thank you for your comment. Please refer to response 11.
Author Response
Authors’ reply/modifications according to the reviewer 1 comments/suggestions (Round 2 – Minor)
General:
The authors sincerely thank the reviewer for their valuable time and thoughtful evaluation of the manuscript in Round 2. We are pleased to note the reviewer’s positive feedback regarding the responses to the majority of comments raised during the previous revision.
Specific response to the reviewer’s suggestions:
Kindly find the attached response to each question one by one:
Point 1: The authors have effectively addressed most of the comments, but there are two points (11 and 14) that are related and concern me. Addressing them correctly will help improve the manuscript. Both refer to citation 37 in the previous manuscript, which is now citation 60. I would like to point out that citation 60 does not exist in the updated version of the manuscript, but rather citation 11. Furthermore, I would like to insist that the authors should enrich the manuscript with recent systematic reviews on the role of the caregiver, stress, and psychosocial well-being, as well as assessment tools for person-centered care and resilience. This would greatly help to improve the quality of the manuscript's content.
Response 1: Thanks for the comment. We apologize for the confusion caused by the reference numbering in our previous response. In the revised version, we have strengthened the theoretical framework in the Introduction and Discussion by incorporating multiple recent systematic reviews addressing caregiver burden, caregiver stress, psychosocial well-being, and related assessment approaches. The revised manuscript now contains a total of 58 references, including several systematic reviews (e.g., references 4, 11, 12, 16, 23, 25, 47, 52, and 55). We believe these additions adequately address Points 11 and 14 (earlier comments) and enhance the scientific relevance of the manuscript.
The authors thank the reviewer once again for the positive and constructive comments.
Reviewer 2 Report
Comments and Suggestions for Authors
The authors have addressed my comments properly.
Author Response
Dear reviewer,
We thank the reviewer once again for their valuable time and thoughtful comments. The authors are pleased that the revisions have satisfactorily addressed the reviewer’s concerns and strengthened the manuscript.
Reviewer 3 Report
Comments and Suggestions for Authors
I would like to express my sincere appreciation to the authors for their meticulous review and faithful revision of the manuscript based on the comments from the first round of review. In particular, the inclusion of the Exploratory Factor Analysis (EFA) results has significantly secured the structural validity of the research instrument. Furthermore, the reinforcement of policy implications, tailored to the specific context of Saudi Arabia , has greatly enhanced the practical value of this paper. Overall, the completeness of the manuscript has improved substantially, and I applaud the authors for their hard work.
However, during the review of the revised manuscript, a few notation errors related to data accuracy were identified. While these appear to be simple mistakes that occurred during the revision and editing process, they are critical issues that could cause confusion for readers. Therefore, it is necessary to correct them before final publication. I would like to provide specific details regarding these points below.
First, I kindly request that you re-verify the sum of the "Patients' Age" distribution presented in Table 1. Currently, the table lists 101 individuals aged less than 20, 136 individuals aged 20 to 40, and 105 individuals aged more than 40. Summing these figures yields 342, which exceeds the total study population of 330. Given that the initial manuscript listed the "more than 40" group as 93, it appears there may have been a transcription error during the revision process. Please check the raw data and correct the figures to ensure accuracy.
Next, corrections are required in the descriptive statistics presented in Table 2. The median for the "Time and social limitations" domain is currently recorded as '2'. Considering that the mean for this domain is 11.29 and the Interquartile Range (IQR) is 8–15, a median of '2' is statistically improbable. It seems highly likely that the leading digit was omitted, and the intended value was '12'; please verify this. Additionally, in the same table, the median notation for the "Interpersonal relationships" domain includes an unnecessary character, 'J'. Please remove this to ensure the table is presented clearly.
Finally, a minor typographical error was found in the citation formatting within the Introduction section. There is an instance currently formatted as "[28-301." where it appears the closing bracket was mistyped as the number '1'. Please correct this to the standard format, such as "[28-30]".
With the exception of the aforementioned notation errors, this manuscript is written in a highly logical and systematic manner, from the research design to the conclusion. I am confident that once the authors address these points, this study will become an excellent contribution that provides readers with trustworthy information.
Author Response
Authors’ reply/modifications according to the reviewer 3 comments/suggestions (Round 2 – Minor revision)
General:
The authors sincerely thank the reviewer for the time and effort spent on evaluating the revised manuscript. We greatly appreciate the constructive feedback and encouraging remarks, which have been invaluable in improving the clarity, methodological rigor, and overall quality of the manuscript.
Specific response to the reviewer’s suggestions:
Kindly find the attached response to each question one by one:
Point 1: I would like to express my sincere appreciation to the authors for their meticulous review and faithful revision of the manuscript based on the comments from the first round of review. In particular, the inclusion of the Exploratory Factor Analysis (EFA) results has significantly secured the structural validity of the research instrument. Furthermore, the reinforcement of policy implications, tailored to the specific context of Saudi Arabia , has greatly enhanced the practical value of this paper. Overall, the completeness of the manuscript has improved substantially, and I applaud the authors for their hard work.
Response 1: Thanks for the comments. he authors are pleased that the inclusion of the Exploratory Factor Analysis (EFA) was found to strengthen the structural validity of the research instrument and that the Saudi-context–specific policy implications were considered to enhance the practical value of the manuscript. We greatly appreciate the reviewer’s thoughtful evaluation and kind recognition of our efforts to improve the completeness and quality of the study.
Point 2: However, during the review of the revised manuscript, a few notation errors related to data accuracy were identified. While these appear to be simple mistakes that occurred during the revision and editing process, they are critical issues that could cause confusion for readers. Therefore, it is necessary to correct them before final publication. I would like to provide specific details regarding these points below.
First, I kindly request that you re-verify the sum of the "Patients' Age" distribution presented in Table 1. Currently, the table lists 101 individuals aged less than 20, 136 individuals aged 20 to 40, and 105 individuals aged more than 40. Summing these figures yields 342, which exceeds the total study population of 330. Given that the initial manuscript listed the "more than 40" group as 93, it appears there may have been a transcription error during the revision process. Please check the raw data and correct the figures to ensure accuracy.
Next, corrections are required in the descriptive statistics presented in Table 2. The median for the "Time and social limitations" domain is currently recorded as '2'. Considering that the mean for this domain is 11.29 and the Interquartile Range (IQR) is 8–15, a median of '2' is statistically improbable. It seems highly likely that the leading digit was omitted, and the intended value was '12'; please verify this. Additionally, in the same table, the median notation for the "Interpersonal relationships" domain includes an unnecessary character, 'J'. Please remove this to ensure the table is presented clearly
Response 2: Thanks for the comments. According to the reviewer’s comments, we rechecked all the revised tables (including Table 1) and corrected a minor percentage rounding error in Table 1; all frequencies and descriptive statistics are now internally consistent (Table 2).
Point 3: Finally, a minor typographical error was found in the citation formatting within the Introduction section. There is an instance currently formatted as "[28-301." where it appears the closing bracket was mistyped as the number '1'. Please correct this to the standard format, such as "[28-30]".
Response 3: Thanks for the comments. We carefully rechecked the citation formatting in the Introduction section and confirmed that the reference range is now correctly presented as “[28–30]” in the revised manuscript. We appreciate the reviewer’s attention to this detail.
Point 4: With the exception of the aforementioned notation errors, this manuscript is written in a highly logical and systematic manner, from the research design to the conclusion. I am confident that once the authors address these points, this study will become an excellent contribution that provides readers with trustworthy information.
Response 4: Thanks for the comments. We sincerely thank the reviewer for the encouraging and thoughtful comments. We are pleased to know that the manuscript is considered logical and systematic, and we appreciate the reviewer’s confidence in the contribution of this study. All notation errors have been carefully addressed, and we believe the revised manuscript now provides clear and reliable information for readers.
The authors thank the reviewer once again for the positive and constructive comments.